

# Effect of donor GSTM3 rs7483 genetic variant on tacrolimus elimination in the early period after liver transplantation

Tao Zhang[1,*], Xiaorong Chen[1,*], Yuan Liu[2] and Lei Zhang[1]

[1] Department of Organ Transplantation, The Third Affiliated Hospital of Sun Yat-sen University, Guangzhou, China
[2] Department of General Surgery, Shanghai General Hospital, Shanghai Jiao Tong University School of Medicine, Shanghai, China
[*] These authors contributed equally to this work.

## ABSTRACT

**Purpose.** Glutathione S-transferase mu (GSTM) belongs to the group of phase II drug-metabolizing enzymes, and the GSTM1 genetic variant has been reported to have a potential association with the metabolism of immunosuppressive drug after renal transplantation. The effect of donor and recipient GSTMs genetic variants on tacrolimus (Tac) metabolism was the focus of our investigation in this study.

**Methods.** A total of 203 liver transplant patients were recruited for the study. In the training set ($n = 110$), twenty-one SNPs in five genes (GSTM1-5) were genotyped by the drug-metabolizing enzymes and transporter (DMET) microarray. CYP3A5 rs776746 and GSTM3 rs7483 were genotyped using a Mass ARRAY platform in the validating set ($n = 93$).

**Results.** Tac C/D ratios of donor GSTM3 rs7483 AA carriers were significantly lower than those with the G allele at weeks 1, 2, 3 and 4 after liver transplantation (LT). Multivariate analysis was conducted on the training set and validating set, donor and recipient CYP3A5 rs776746, donor GSTM3 rs7483 and total bilirubin were identified as independent predictors of Tac C/D ratios in the early period after LT. Combining CYP3A5 rs776746 and donor GSTM3 rs7483 genotypes, Tac C/D ratios were observed to be increasingly lower with increasing numbers of alleles associated with fast metabolism. Moreover, the risk of a supratherapeutic $C_0$ (Tac > 15 ug/L) was significantly higher for poor metabolizers than the other groups at week 1 after LT.

**Conclusions.** There was a significant association between the donor GSTM3 rs7483 genetic variant and Tac metabolism in the early period after LT. Genotype classification might have a better predictive ability of the initial Tac doses.

## INTRODUCTION

Tacrolimus (Tac) is a pivotal immunosuppressive agent for the prevention of allograft rejection in liver transplantation (LT). Posttransplant survival has greatly improved with its use. However, the clinical use of Tac is complicated due to its narrow therapeutic index and the large inter- and intra-individual variation in its pharmacokinetics

Corresponding authors
Yuan Liu, paradisetong@163.com
Lei Zhang,
zhanglei19820612@163.com

(*Virunya et al., 2024*; *Siqi et al., 2024*). Underdosing of Tac may cause underimmuno-suppression and acute graft rejection, whereas overdosing puts patients at the risk of serious post-transplant complications, including infection, nephrotoxicity, neurotoxicity and diabetes mellitus (*Oren et al., 2024*; *Azhie et al., 2021*; *Pageaux et al., 2009*). Therefore, there has been increasing recognition of the need for means of individualizing Tac treatment to rapidly achieve target blood concentration in the early period after LT.

Pharmacogenetics has the potential to explain 20%–95% of the inter- and intra-individual variation observed in drug metabolism and treatment response (*Evans & McLeod, 2003*). Individual single nucleotide polymorphisms (SNPs) may change the expression or biologic activity of protein that has a physiological effect on the organism (*Siqi et al., 2024*). Tac is a substrate of metabolic and transport enzymes, the genetic variants of cytochrome P-450 3A4/5 (CYP3A4/5), multidrug resistance protein 1 (ABCB1) could partly alter the metabolism and clearance of Tac in organ transplantation (*Liu et al., 2022*; *Aouam et al., 2015*; *Liu et al., 2013*). However, the inter- and intra-individual variation of Tac pharmacokinetics could not be fully explained by these SNPs, and additional determinants remain to be uncovered.

Glutathione S-transferase mu (GSTM) are some of the most abundant GSTs found in the human liver and brain (*Uno et al., 2020*), and belong to the group of phase II drug-metabolising enzymes that facilitate the detoxification of toxic chemical, therapeutic drugs and products of oxidative stress (*Doerksen et al., 2023*; *Uno et al., 2020*). There are five distinct human isoforms in the GSTM cluster (GSTM1, GSTM2, GSTM3, GSTM4, GSTM5). In addition, a number of polymorphisms have been reported and characterized in these genes (*Santos et al., 2019*; *Rodríguez et al., 2018*; *Tatemichi et al., 2009*). The genetic variants of these GSTMs have the potential to alter an individual's susceptibility to carcinogens and toxins, and influence the toxicity and efficacy of drug treatment. And a previous study has reported that the polymorphisms of GSTM1 had a potential association with the elimination of Tac in renal transplant recipients (*Hayes & Strange, 2000*; *Singh et al., 2009*; *Kearns et al., 2002*).

The current study was based on our drug-metabolizing enzymes and transporter (DMET) microarray, which contains 1936 genetic variants in 225 related genes. Firstly, we investigated the association between GSTMs SNPs within DEMT microarray and Tac pharmacokinetics from our study subjects, the donors and recipients of 110 liver transplantations (Cohort A). These genetic variants included GSTM2 (rs530021), GSTM3 (rs7483, rs4646412), GSTM4 (rs506008), GSTM5 (rs1296954, rs11807). The significant markers were then validated in the test set (Cohort B: $n = 93$).

## PATIENTS & METHODS

### Patients

From Jan 2017 and Dec 2020, we meticulously screened patients who underwent orthotopic liver transplantation at The Third Affiliated Hospital of Sun Yat-sen University, China (Cohort A = 110, Cohort $B = 93$.) The criteria were: (1) age $\geq$18 years, (2) LT from DCD/DBD, (3) immunosuppressive regimen was triple therapy incorporating
**Table 1 Baseline demographic characteristics.**

| Demographic variables | Total population ($n = 203$) | |
| --- | --- | --- |
| | Cohort A ($n = 110$) | Cohort B ($n = 93$) |
| Recipient age (Years) | 47.48 ± 9.20 | 48.85 ± 10.86 |
| Recipient gender (male/female, n) | | |
| Child-pugh score | 91 (82.7%) /19 (17.3%) | 74 (79.6%)/19 (20.4%) |
| Primary disease (n) | 7.12 ± 2.12 | 9.38 ± 2.44 |
|    HBV cirrhosis | | |
|    HCV cirrhosis | | |
|    Any with HCC | 29 (26.4%) | 58 (62.3%) |
|    Autoimmune cirrhosis | – | 1 (1.1%) |
|    Alcoholic cirrhosis | 69 (62.7%) | 25 (26.9%) |
|    Primary biliary cirrhosis | 4 (3.6%) | 1 (1.1%) |
|    Wilson Disease | 2 (1.8%) | 1 (1.1%) |
|    Schistosomal cirrhosis | 3 (2.7%) | 4 (4.3%) |
|    Budd-Chiari syndrome | 2 (1.8%) | 2 (2.1%) |
| | 1 (1.0%) | – |
| | – | 1 (1.1%) |

tacrolimus, mycophenolate, and steroid, (4) Signed informed consent. The exclusion criteria were: (1) multiorgan transplant patients, (2) follow-up time less than 1 month, (3) Immunosuppressive regimen altered (*e.g.*, to cyclosporin). The patient characteristics were summarized in Table 1.

## Ethics statement

Human participants written informed consent was obtained from the subjects or the next-of-kin. The retrospective research was approved by the Ethics Committee of The Third Affiliated Hospital of Sun Yat-sen University (A2023-232-01). All procedures were performed in accordance with the 1964 Declaration of Helsinki and its later amendments or comparable ethical standards. Assurances were made to ensure that no livers were obtained from the executed prisoners (1) Voluntary organ donation (DCD/DBD), (2) human participants written informed consent was obtained from the next-of-kin of donors, (3) the quality of donor's liver meets the transplantation standard, (4) China made posthumous voluntary donation the only legitimate source of organs in 2015).

## Data collection

Therapeutic drug monitoring (TDM) was performed routinely after LT. The $C_0$ was measured in laboratories by the Pro-TracTMII tacrolimus ELISA kit (Diasorin) with a microparticle enzyme immunoassay (ELx 800NB analyzer; BioTek), using the whole blood collected before the morning administration. We used the Tac C/D ratio (ug/L per mg/kg) as a measure of Tac pharmacokinetics, calculating it by dividing the trough concentration (ug/L) by the dosage adjusted for body weight (mg/kg). Clinical data encompassed demographic insights (age, gender, child-pugh score, primary liver disease), liver function index (ALT, AST, TB, DB, Alb), and renal function index (Cr, Urea).

## Genomic DNA isolation & genotyping

Genomic DNA obtained from donor and recipient hepatic samples was purified using the AllPrep DNA/RNA mini-kit (Qiagen, Hilden, Germany), previously secured at −80 °C. In the training set (Cohort A), genomic DNA from 110 patients was genotyped by Affymetrix DMET Plus array according to the molecular inversion probe (MIP) technology as previously described (*Dumaual et al., 2007*; *Di Martino et al., 2011a*; *Di Martino et al., 2011b*). And in the validated set (Cohort B), genotyping of CYP3A5 rs776746 and GSTM3 rs7483 was conducted using the Sequenom MassARRAY SNP genotyping platform (Sequenom, San Diego, CA, USA) (*Gabriel, Ziaugra & Tabbaa, 2009*). The sequencing primers for rs776746 and rs7483 were as follows: rs776746: forward 5′-AGGAAGCCAGACTTTGATCATTATGTT-3′, reverse 5′-GAGAGTGGCATAGGAGATACCCA-3′; rs7483: forward 5′-CCAGTATCGCAGCGATTCAATT-3′, reverse 5′-GCCTACTTACAGTCTGATCAGTTCTG-3′.

## Statistical analysis

SPSS version 19.0 (SPSS, Chicago, IL, USA) was used for statistical analysis. Genetic equilibrium and allele distribution were analyzed using the SHEsis software platform (*Shi & He, 2005*). Tac C/D ratios were assessed for normality of distribution and logarithmically transformed if non-normal. Mean substitution where we substituted the missing values. Tac C/D ratios between genotype groups were analyzed using Student independent $t$-tests or Mann–Whitney test. Given the null hypothesis that the group means are equal, the ANOVAs were conducted. We utilized stepwise multiple linear regression to evaluate the impact of the CYP3A5 rs776746 and GSTM3 rs7483 genetic variant on Tac C/D ratios, along with clinical characteristics such as ALT, AST, TB, DB, Alb, Cr, Urea. Variables with a univariate $p < 0.10$ were included in multivariate analysis. The enter method was used to confirm the results derived from the training dataset. We assessed the impact of genotype cluster on the risk of Tac blood concentrations >15 ug/L using logistic regression analysis. Statistical significance was determined by two-tailed $p$-values of less than 0.05.

# RESULTS

## Gene distribution

The distribution of genotypes of CYP3A5 rs776746 and GSTM1-5 SNPs within the DEMT microarray is presented in Table S1. Fourteen SNPs with no variants in our study population were excluded for further analysis. The remaining seven SNPs conformed to Hardy–Weinberg equilibrium ($P > 0.05$). No significant linkage disequilibrium was observed between the CYP3A5 rs776746 and the six GSTM2-5 SNPs.

## Influence of CYP3A5 and GSTM2-5 genetic vatiants on Tac elimination (Cohort A)

The association between donor CYP3A5 and GSTM2-5 SNPs and Tac C/D ratios in the early period after LT was shown in Table 2. Among CYP3A5 rs776746 carriers, those with AA/AG genotypes have been observed to have lower Tac C/D ratios than GG genotype carriers at weeks 1, 2, 3, 4 ($p = 0.005, 0.007, 0.002, <0.001$, respectively). Tac C/D ratios

**Table 2  Tac C/D ratios according to donor CYP3A5 and GSTMs genotypes after drug initiation (Cohort A, $n = 110$).**

| Gene | SNP | Genotype | Week 1 | | Week 2 | | Week 3 | | Week 4 | |
|------|-----|----------|--------|---|--------|---|--------|---|--------|---|
| | | | C/D ratios | P | C/D ratios | P | C/D ratios | P | C/D ratios | P |
| CYP3A5 | rs776746 | AA+AG | $215.0 \pm 140.3$ | 0.005 | $139.1 \pm 97.0$ | 0.007 | $129.6 \pm 120.3$ | 0.002 | $114.7 \pm 82.9$ | <0.001 |
| | | GG | $348.9 \pm 253.2$ | | $180.8 \pm 135.2$ | | $203.0 \pm 203.0$ | | $229.4 \pm 273.4$ | |
| GSTM2 | rs530021 | CC | $265.0 \pm 189.3$ | 0.739 | $159.2 \pm 116.5$ | 0.498 | $168.0 \pm 171.9$ | 0.464 | $176.5 \pm 218.9$ | 0.256 |
| | | CG+GG | $341.9 \pm 301.4$ | | $157.4 \pm 130.6$ | | $142.3 \pm 144.5$ | | $135.0 \pm 111.5$ | |
| GSTM3 | rs7483 | AA | $231.0 \pm 164.9$ | 0.035 | $127.3 \pm 73.6$ | 0.010 | $120.4 \pm 82.4$ | 0.035 | $116.1 \pm 71.1$ | 0.002 |
| | | AG+GG | $328.2 \pm 243.6$ | | $195.1 \pm 146.6$ | | $213.4 \pm 219.6$ | | $235.20 \pm 180.3$ | |
| | rs4646412 | GG | $277.7 \pm 215.8$ | 0.822 | $154.5 \pm 115.8$ | 0.096 | $162.8 \pm 167.1$ | 0.522 | $171.0 \pm 214.1$ | 0.174 |
| | | GT+TT | $267.8 \pm 151.0$ | | $213.3 \pm 149.3$ | | $176.3 \pm 181.0$ | | $163.2 \pm 85.7$ | |
| GSTM4 | rs506008 | GG | $263.8 \pm 193.7$ | 0.363 | $157.4 \pm 116.3$ | 0.946 | $165.1 \pm 170.0$ | 0.868 | $173.1 \pm 216.2$ | 0.652 |
| | | GA+AA | $367.4 \pm 297.2$ | | $170.0 \pm 134.7$ | | $156.2 \pm 155.1$ | | $150.2 \pm 117.2$ | |
| GSTM5 | rs1296954 | GG | $272.8 \pm 189.2$ | 0.764 | $155.4 \pm 116.2$ | 0.720 | $144.2 \pm 146.3$ | 0.149 | $130.8 \pm 82.4$ | 0.089 |
| | | AG+AA | $284.3 \pm 248.3$ | | $165.7 \pm 123.0$ | | $202.8 \pm 199.5$ | | $255.6 \pm 334.5$ | |
| | rs11807 | AA | $265.5 \pm 203.5$ | 0.526 | $148.9 \pm 91.2$ | 0.797 | $166.1 \pm 158.9$ | 0.276 | $177.5 \pm 225.7$ | 0.731 |
| | | AG+GG | $294.9 \pm 222.4$ | | $175.3 \pm 151.9$ | | $160.3 \pm 182.5$ | | $158.5 \pm 172.7$ | |

of donor GSTM3 rs7483 AA genotype were $231.0 \pm 164.9$, $127.3 \pm 73.6$, $120.4 \pm 82.4$ and $116.1 \pm 71.1$ at weeks 1, 2, 3 and 4 respectively. For AG and GG genotype carriers, the corresponding Tac C/D ratios at each time point were $328.2 \pm 243.6$, $195.1 \pm 146.6$, $213.4 \pm 219.6$ and $235.20 \pm 180.3$. The differences were significant ($p = 0.035$, 0.010, 0.035, 0.002, respectively).

The effects of recipient CYP3A5 and GSTM2-5 SNPs on Tac C/D ratios in the early period after LT was shown in Table 3. Tac C/D ratios of recipient CYP3A5 rs776746 AA/AG carriers were $220.7 \pm 190.0$, $126.2 \pm 77.1$, $120.5 \pm 107.3$ and $141.7 \pm 222.3$ at week 1, 2, 3 and 4 respectively, and $328.8 \pm 216.7$, $188.7 \pm 139.9$, $202.0 \pm 199.6$ and $195.2 \pm 190.5$ for GG genotype carriers. Tac C/D ratios of recipient CYP3A5 rs776746 AA/AG carriers were significantly lower than GG carriers at all investigated time points ($p = 0.001$, 0.006, 0.005, 0.003, respectively). However, there was not significant association between the recipient GSTM2-5 genotype groups in the early post-transplantation period.

## Multivariate analysis for factors influencing Tac metabolism (Cohort A and B)

We investigated the effect of the genetic and clinical factors on Tac metabolism in the early post-transplantation period through multiple linear regression analysis. The cofactors that were incorporated into the analysis included CYP3A5 rs776746 and donor GSTM3 rs7483, liver function indices (ALT, AST, TB, DB, Alb), and renal function indices (Cr and Urea). In the training set (Cohort A: Table 4), donor and recipient CYP3A5 rs776746, donor GSTM3 rs7483 and the liver function indices (TB, DB) were identified as independent predictors of Tac metabolism in the early period after LT.

To confirm the effect of significant factors on Tac metabolism, we proceeded with further analysis in the validating set (Cohort B: Table 5). Donor and recipient CYP3A5 rs776746 and donor GSTM3 rs7483 were significantly associated with Tac elimination in

**Table 3  Tac C/D ratios according to recipient CYP3A5 and GSTMs genotypes after drug initiation (Cohort A, $n = 110$).**

| Gene | SNP | Genotype (n) | Week 1 | | Week 2 | | Week 3 | | Week 4 | |
|---|---|---|---|---|---|---|---|---|---|---|
| | | | C/D ratios | P | C/D ratios | P | C/D ratios | P | C/D ratios | P |
| CYP3A5 | rs776746 | AA+AG (58) | 220.7 ± 190.0 | 0.001 | 126.2 ± 77.1 | 0.006 | 120.5 ± 107.3 | 0.005 | 141.7 ± 222.3 | 0.003 |
| | | GG (52) | 328.8 ± 216.7 | | 188.7 ± 139.9 | | 202.0 ± 199.6 | | 195.2 ± 190.5 | |
| GSTM2 | rs530021 | CC (93) | 275.8 ± 215.3 | 0.729 | 158.9 ± 115.5 | 0.392 | 166.1 ± 168.6 | 0.424 | 174.2 ± 215.2 | 0.570 |
| | | CG+GG (17) | 284.5 ± 175.7 | | 159.2 ± 141.9 | | 146.4 ± 164.1 | | 135.1 ± 99.9 | |
| GSTM3 | rs7483 | AA (59) | 276.4 ± 232.5 | 0.519 | 155.4 ± 117.8 | 0.508 | 177.1 ± 190.6 | 0.591 | 177.5 ± 242.4 | 0.383 |
| | | AG+GG (51) | 277.2 ± 183.7 | | 163.2 ± 119.5 | | 148.2 ± 135.5 | | 162.2 ± 158.2 | |
| | rs4646412 | GG (101) | 274.3 ± 207.4 | 0.855 | 159.4 ± 121.9 | 0.687 | 169.1 ± 177.3 | 0.689 | 175.7 ± 218.6 | 0.908 |
| | | GT+TT (9) | 296.2 ± 241.6 | | 155.2 ± 86.4 | | 126.3 ± 49.3 | | 130.8 ± 64.1 | |
| GSTM4 | rs506008 | GG (96) | 273.7 ± 215.1 | 0.419 | 157.9 ± 115.3 | 0.615 | 165.0 ± 168.1 | 0.689 | 174.2 ± 215.2 | 0.570 |
| | | GA+AA (14) | 303.1 ± 171.4 | | 167.5 ± 145.7 | | 154.6 ± 169.6 | | 135.1 ± 100.0 | |
| GSTM5 | rs1296954 | GG (73) | 303.2 ± 229.4 | 0.153 | 157.7 ± 96.9 | 0.429 | 175.9 ± 176.8 | 0.253 | 180.7 ± 231.0 | 0.540 |
| | | AG+AA (37) | 240.8 ± 177.6 | | 160.8 ± 145.4 | | 146.0 ± 152.8 | | 155.8 ± 158.1 | |
| | rs11807 | AA (67) | 271.6 ± 201.8 | 0.916 | 172.2 ± 132.1 | 0.214 | 169.8 ± 177.1 | 0.662 | 159.6 ± 170.2 | 0.490 |
| | | AG+GG (43) | 288.6 ± 231.6 | | 130.0 ± 73.2 | | 148.9 ± 141.6 | | 198.3 ± 282.4 | |

**Table 4  Multiple linear regression model for log-transformed Tac C/D ratios in the first month (Cohort A, $n = 110$, stepwise method).**

| | | B | Beta | T | Sig. | VIF | Adjusted $R^2$ | D-W |
|---|---|---|---|---|---|---|---|---|
| Week1 | (Constant) | 1.406 | | 8.970 | .000 | | 0.248 | 1.806 |
| | Donor rs7483 | .157 | .225 | 2.574 | .012 | 1.108 | | |
| | Donor rs776746 | .225 | .322 | 3.708 | .000 | 1.006 | | |
| | Recipient rs776746 | .232 | .334 | 3.827 | .000 | 1.013 | | |
| Week2 | (Constant) | 1.421 | | 11.476 | .000 | | 0.268 | 1.897 |
| | Total bilirubin | .001 | .291 | 3.376 | .001 | 1.055 | | |
| | Donor rs7483 | .119 | .210 | 2.416 | .018 | 1.072 | | |
| | Donor rs776746 | .137 | .242 | 2.876 | .005 | 1.005 | | |
| | Recipient rs776746 | .158 | .279 | 3.297 | .001 | 1.018 | | |
| Week3 | (Constant) | 1.332 | | 9.962 | .000 | | 0.245 | 2.107 |
| | Direct bilirubin | .003 | .317 | 3.681 | .007 | 1.038 | | |
| | Donor rs7483 | .121 | .195 | 2.216 | .029 | 1.053 | | |
| | Donor rs776746 | .196 | .317 | 3.681 | .000 | 1.008 | | |
| | Recipient rs776746 | .148 | .239 | 2.772 | .007 | 1.013 | | |
| Week4 | (Constant) | 1.139 | | 8.346 | .000 | | 0.357 | 2.163 |
| | Direct bilirubin | .002 | .176 | 2.108 | .038 | 1.047 | | |
| | Donor rs7483 | .202 | .323 | 3.884 | .000 | 1.045 | | |
| | Donor rs776746 | .279 | .448 | 5.463 | .000 | 1.013 | | |
| | Recipient rs776746 | .142 | .227 | 2.766 | .007 | 1.013 | | |

**Table 5   Multiple linear regression model for log-transformed Tac C/D ratios in the first month (Cohort B, $n = 93$, enter method).**

| | | B | Beta | T | Sig. | VIF | Adjusted $R^2$ | D-W |
|---|---|---|---|---|---|---|---|---|
| Week1 | (Constant) | 1.772 | | 9.072 | .000 | | 0.170 | 1.559 |
| | Total bilirubin | .001 | .249 | 2.231 | .029 | 1.067 | | |
| | Donor rs7483 | .108 | .208 | 2.083 | .040 | 1.042 | | |
| | Recipient rs776746 | .194 | .316 | 2.871 | .005 | 1.037 | | |
| Week2 | (Constant) | 1.805 | | 8.524 | .000 | | 0.084 | 1.173 |
| | Donor rs776746 | .219 | .322 | 3.007 | .004 | 1.035 | | |
| Week3 | (Constant) | 1.434 | | 8.894 | .000 | | 0.207 | 1.743 |
| | Donor rs7483 | .140 | .238 | 2.343 | .022 | 1.018 | | |
| | Donor rs776746 | .193 | .329 | 3.199 | .002 | 1.039 | | |
| | Recipient rs776746 | .171 | .285 | 2.788 | .007 | 1.031 | | |
| Week4 | (Constant) | 1.410 | | 7.583 | .000 | | 0.176 | 1.692 |
| | Donor rs7483 | .161 | .265 | 2.363 | .021 | 1.020 | | |
| | Donor rs776746 | .170 | .279 | 2.495 | .015 | 1.015 | | |
| | Recipient rs776746 | .153 | .246 | 2.204 | .031 | 1.014 | | |

**Table 6   Combined analysis of donor CYP3A5 rs776746 allele A, recipient CYP3A5 rs776746 allele A and donor GSTM3 rs7483 allele A on Tac C/D ratios after drug initiation ($n = 203$).**

| | | Num[a] | | | |
|---|---|---|---|---|---|
| | | 0–1 (Group 1, $n = 31$) | 2–3 (Group 2, $n = 131$) | 4–6 (Group 3, $n = 41$) | $p$-value |
| Month 1 | Week 1 | 314.24 ± 259.76 | 224.17 ± 166.19 | 144.11 ± 126.04 | 0.001 |
| | Week 2 | 268.46 ± 182.46 | 172.60 ± 155.42 | 148.13 ± 132.98 | 0.004 |
| | Week 3 | 296.54 ± 216.00 | 158.52 ± 135.54 | 115.14 ± 70.94 | <0.001 |
| | Week 4 | 305.84 ± 242.48 | 153.84 ± 170.85 | 95.91 ± 70.07 | <0.001 |

**Notes.**
[a]The number of alleles associated with fast metabolism.

the early period after LT. In addition, total bilirubin was also a predictor of Tac elimination for week 1.

### Combined effects of CYP3A5 rs776746 and GSTM3 rs7483 genotypes

Donor and recipient CYP3A5 rs776746 allele A and donor GSTM3 rs7483 allele A were associated with fast Tac metabolism as stated above, we combined CYP3A5 rs776746 and GSTM3 rs7483 genotypes and investigated the effects of the number of alleles associated with fast metabolism on Tac C/D ratios (Table 6). Group 1 consisted of 0-1 alleles (poor metabolizers); group 2 contained 2-3 alleles (intermediate metabolizers); group 3 contained 4-6 alleles (extensive metabolizers). With increasing numbers of alleles associated with fast metabolism, Tac C/D ratios were increasingly lower at each time points within the first month (group 1 >group 2 >group 3; $p = 0.001$, 0.004, <0.001, <0.001, respectively).

The geometric mean of Tac concentrations were 13.6 ug/L, 9.2 ug/L, 5.7 ug/L at week 1 for patients from group 1, group 2 and group 3, respectively. Logistic regression analysis showed that the risk of presenting a supratherapeutic $C_0$ (Tac >15 ug/L) at week 1 was significantly higher for group 1 (Fig. 1), compared with group 2 (odds ratio: 4.143; 95%
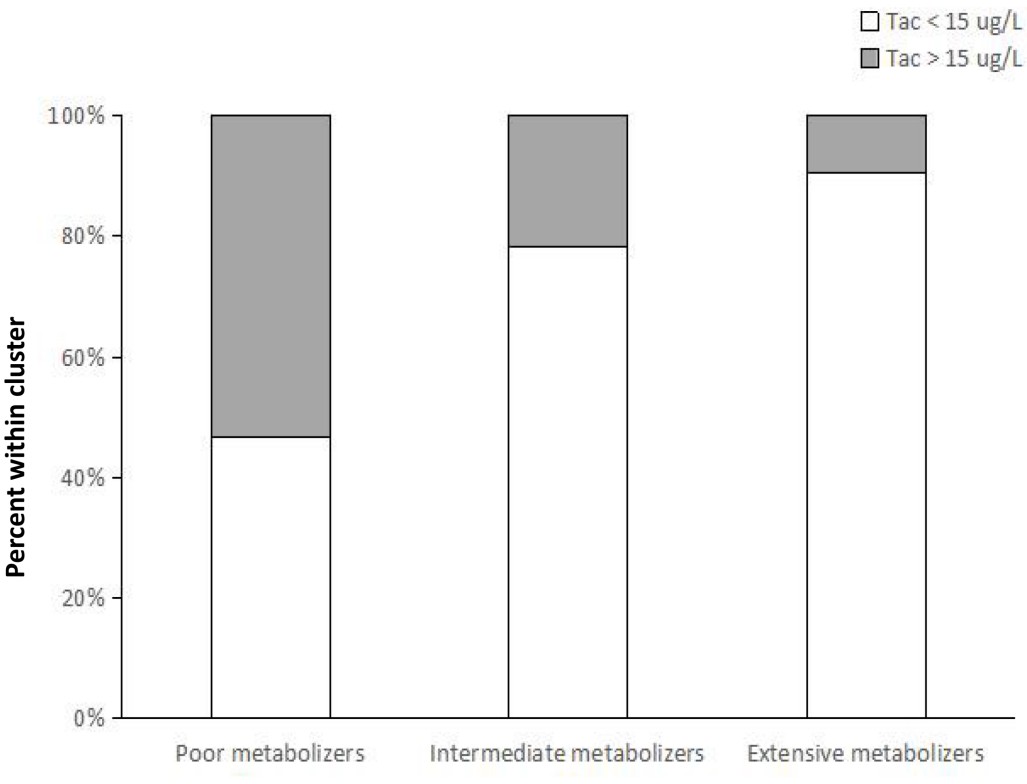

**CYP3A5 and GSTM3 genotype clusters**

**Figure 1** **Percentage of patients within each metabolizer cluster stratified by values of $C_0$ below or above the 15-ug/L supratherapeutic threshold at week 1 after liver transplantation.** The risk of presenting a supratherapeutic $C_0$ (Tac > 15 ug/L) at week 1 was significantly higher for poor metabolizers, compared with intermediate metabolizers ($p = 0.016$) and extensive metabolizers ($p = 0.008$).

CI [1.305–13.175]; $p = 0.016$) and group 3 (odds ratio: 3.295 ; 95% CI [1.356–8.005]; $p = 0.008$). However, no significant differences were observed between the different groups regarding to the risk of a $C_0$ <8 ug/L ($p = 0.742, 0.163$, respectively). These results indicated that poor metabolizers require lower Tac doses to reach the target blood concentrations and genotype classification demonstrated a better predictive ability for the initial Tac doses after LT.

## DISCUSSION

The study is the first time to investigate the effects of the GSTMs genetic variants on Tac metabolism in the early period after LT. In the training set, we found that Tac C/D ratios of donor GSTM3 rs7483 AA carriers were significantly lower than those with the G allele at weeks 1, 2, 3 and 4. No significant association between the other GSTM2-5 genotype groups were observed at all investigated time points. In multiple linear regression analysis, donor GSTM3 rs7483 genetic variant was identified as an independent predictor of Tac elimination in the early period after LT both in the two cohorts. Of 203 liver transplant patients, the distribution of genotypes for the GSTM3 rs7483 genetic variant were 6.7%GG,

40.6%GA, 52.7%AA, aligning with the previous research on the Chinese population (*Tan et al., 2013*; *Tetlow et al., 2004*).

Our results are agreement with the increased function of the GSTM3 rs7483 genetic variant and the expected fast metabolism of Tac. GSTM3 belongs to the phase II drug-metabolising enzymes that plays a key role in the detoxification of chemical agents. Increasing evidences have revealed that the capacity to metabolise drugs may be partly affected by the genetic variants in the population (*Liu et al., 2022*; *Aouam et al., 2015*; *Liu et al., 2013*). A research had reported that the genetic variant of GSTM1 had a potential association with Tac elimination in the first month after transplantation (*Singh et al., 2009*). In addition, the SNP rs7483 (224 G>A) in GSTM3 results in the substitution of valine (Val) for isoleucine (Ile) in the GSTM3 protein, which has been reported to significantly increase the activity of the drug-metabolising enzyme (*Tetlow et al., 2004*; *Shiota et al., 2016*). Therefore, the increased enzymatic activity might affect Tac metabolism.

In the present study, we have further confirmed that patients with CYP3A5 rs776746 AA/AG genotype (expressers) require lower Tac doses to achieve the target blood concentration compared with CYP3A5 GG genotype carriers (nonexpressers) (*Du et al., 2024a*; *Du et al., 2024b*; *Nuchjumroon et al., 2022*; *Dong et al., 2022*; *Sallustio et al., 2021*; *Everton Janaína et al., 2021*; *Kelava et al., 2020*; *Coller et al., 2019*; *Tang et al., 2019*; *Zhang et al., 2018*). The effects of donor and recipient CYP3A5 rs776746 and donor GSTM3 rs7483 SNPs appeared independent, the combined analysis of CYP3A5 rs776746 and donor GSTM3 rs7483 genotypes shown a more significant impact on Tac pharmacokinetics compared to examining the genotypes separately. Tac C/D ratios were significantly lower with increasing numbers of alleles associated with fast metabolism: poor metabolizers (Group 1) >intermediate metabolizers (Group 2) >extensive metabolizers (Group 3). Furthermore, our results demonstrated that the risk of a supratherapeutic $C_0$ (Tac >15 ug/L) at week 1 was significantly higher for poor metabolizers than for intermediate metabolizers and extensive metabolizers. Although therapeutic drug monitoring (TDM) is helpful for subsequent dosage modification, it provides no information for the initial dose. Therefore, genotype classification might help clinicians to individualize the Tac starting dose after LT.

Beyond genetic factors, clinical parameters (total bilirubin, direct bilirubin) were found to be significantly correlated with Tac elimination after LT, which was consistent with the previous study (*Fan et al., 2015*; *Luo et al., 2016*). It is well known that biliary excretion is associated with the elimination of Tac metabolites (*Siqi et al., 2024*), and therefore, alterations in liver function could significantly influence Tac pharmacokinetics.

There were several limitations in our study. Firstly, these results were obtained from a relatively small number of Chinese patients. Confirmation of the effects of the GSTMs SNP is need in larger or more diverse populations. Secondly, this study lack some experimental data to support the clinical observation. Therefore, further clinical and mechanistic studies are needed to elucidate our findings.

In summary, we have demonstrated that donor GSTM3 rs7483 genetic variant was associated with fast Tac metabolism in the early post-transplantation period. Combined CYP3A5 rs776746 and donor GSTM3 rs7483 genotypes could assist in the precise

determination of initial Tac doses to achieve a target concentration and reduce the risk of reaching supratherapeutic concentration.

### Funding
This work was supported by the National Natural Science Foundation of China (82300748). The funders had no role in study design, data collection and analysis, decision to publish, or preparation of the manuscript.

### Grant Disclosures
The following grant information was disclosed by the authors:
The National Natural Science Foundation of China: 82300748.

### Competing Interests
The authors declare there are no competing interests.

### Author Contributions
- Tao Zhang performed the experiments, analyzed the data, prepared figures and/or tables, authored or reviewed drafts of the article, and approved the final draft.
- Xiaorong Chen performed the experiments, analyzed the data, prepared figures and/or tables, and approved the final draft.
- Yuan Liu performed the experiments, analyzed the data, authored or reviewed drafts of the article, and approved the final draft.
- Lei Zhang conceived and designed the experiments, authored or reviewed drafts of the article, and approved the final draft.

### Human Ethics
The following information was supplied relating to ethical approvals (*i.e.*, approving body and any reference numbers):
   The research was approved by the Ethics Committee of The Third Affiliated Hospital of Sun Yat-sen University (A2023-232-01).

### Ethics
The following information was supplied relating to ethical approvals (*i.e.*, approving body and any reference numbers):
   The research was approved by the Ethics Committee of The Third Affiliated Hospital of Sun Yat-sen University (A2023-232-01).

### Microarray Data Deposition
The following information was supplied regarding the deposition of microarray data:
   https://www.biosino.org/node/project/detail/OEP00005561.

### Data Availability
   The raw data are available in the Supplementary Files.

## Supplemental Information

Supplemental information for this article can be found online at http://dx.doi.org/10.7717/peerj.18360#supplemental-information.

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
