# Peer review of "Effect of donor GSTM3 rs7483 genetic variant on tacrolimus elimination in the early period after liver transplantation"

_PeerJ, doi:10.7717/peerj.18360_

## Round 0.1 · original submission · Major Revisions

Dear Dr. Zhang,

If you feel you can revise your manuscript according to the reviewers' comments, please revise your manuscript and submit it. Please also send us your written responses to each of the reviewers' comments.

Yours,

Yoshi

Prof. Yoshinori Marunaka, M.D., Ph.D.

Reviewer 1 ·

Basic reporting

The present study investigated the association of GSTM polymorphisms in both donors and recipients with tacrolimus metabolism in liver transplant recipients, identifying several significant polymorphisms. The manuscript is well-organized, and the language used is suitable.However, there are several major concerns that should be addressed.
1.The literature cited is somewhat outdated.

2.In the Introduction (lines 86-87), the author should provide evidence supporting the potential association between GSTMs and tacrolimus metabolism. Additionally, references 16-18 do not confirm an association between GSTM polymorphism and the elimination of tacrolimus.

3.In the Abstract, within the Background section, the author should mention the potential link between GSTM and tacrolimus.

Experimental design

1. In the Patients and Methods section, lines 109-110, the author needs to clarify whether the study was conducted retrospectively or prospectively since the ethics approval was granted in 2023, although the study commenced in 2017. If it is retrospective, the author should explain how the samples were obtained.

2. In the Statistical Analysis section, lines 152-154, the criteria for the P-value are unclear. Do the authors mean that variables with a P-value less than 0.1 in univariate analysis were selected for multiple regression analysis, and then those with a P-value less than 0.05 in multiple regression analysis were retained in the final model? From Table 4, it appears that the stepwise method was employed for multiple regression analysis. Therefore, the stepwise method should be described more clearly.

3.The raw data shows some missing values. How did the author handle these missing values? This should be clarified in the statistical analysis section.

4.In lines 160-161, does the microarray only contain CYP3A5 rs776746 and GSTM1-5 SNPs? If not, the author needs to explain why only the current SNPs were chosen for reporting.

5.Line 156: Please specify the statistical software used.

Validity of the findings

1. Table 5 presents results for cohort B. Generally, the validating set is used to confirm the results derived from the training dataset rather than conducting another multiple regression analysis.

2.In Table 3, it would be beneficial to provide the numbers of subjects with different genotypes.

Reviewer 2 ·

Basic reporting

The two queues used in this study are from the same center. External validation sets can be added to increase the reliability and generalization of the model

Experimental design

1.This study uses multiple linear regression model to explore the predictors of tacrolimus drug concentration, and should show more indicators at the same time to ensure that the data meet the basic assumptions of linear regression (such as linearity, independence, homovariance and normality)



2. Although multivariable linear regression model is used in the study, the prediction ability evaluation indexes of the model (such as R ², adjusted R ², etc.) are not provided, which are helpful to understand the prediction effect of the model on new data.

Validity of the findings

The research results need more biological explanations, especially the specific mechanism of how GSTM3 rs7483 gene mutation affects tacrolimus metabolism.

Additional comments

The manuscript needs extensive revision before publication

---

## Round 0.2 · Minor Revisions

Please address these final minor changes and resubmit.

Yours,
Yoshi
Prof. Yoshinori Marunaka, M.D., Ph.D.

Reviewer 1 ·

Basic reporting

Thank you for the authors' response. After carefully reviewing reference 17 again, I noticed a significant difference in the daily dosage of tacrolimus. However, no notable difference was observed in dose-adjusted levels of Tac, which raises doubts about the association between the polymorphism and Tac elimination. While it's possible that the small sample size may have contributed to this, the statement in the manuscript that "a previous study reported a significant association between GSTM1 polymorphisms and Tac elimination in renal transplant recipients" seems too definitive. I recommend that the author consider expressing this in a more precise manner.
According to the above comment, I suggest revising the first sentence in the abstract to “Glutathione S-transferase mu (GSTM) belongs to the group of phase II drug-metabolizing enzymes, and the GSTM1genetic variant has been reported to have a potential association with the metabolism of immunosuppressive drug after renal transplantation”.

Experimental design

No further comment.

Validity of the findings

No further comment.

Additional comments

No further comment.

---

## Round 0.3 · accepted · Accept

Congratulations.
Yours,
Yoshi
Prof. Yoshinori Marunaka, M.D., Ph.D.

Reviewer 1 ·

Basic reporting

no comment

Experimental design

no comment

Validity of the findings

no comment

Additional comments

no comment